# The Design of a Novel 2-42 GHz MEMS True-Time Delay Network for Wideband Phased Array Systems

**DOI:** 10.3390/mi14020246

**Published:** 2023-01-18

**Authors:** Qiannan Wu, Zemin Shi, Xudong Gao, Jing Li, Yongxin Zhan, Guangzhou Zhu, Junqiang Wang, Mengwei Li

**Affiliations:** 1School of Semiconductors and Physics, North University of China, Taiyuan 030051, China; 2Academy for Advanced Interdisciplinary Research, North University of China, Taiyuan 030051, China; 3Center for Microsystem Intergration, North University of China, Taiyuan 030051, China; 4School of Instrument and Intelligent Future Technology, North University of China, Taiyuan 030051, China; 5School of Instrument and Electronics, North University of China, Taiyuan 030051, China; 6Key Laboratory of Dynamic Measurement Technology, North University of China, Taiyuan 030051, China

**Keywords:** true-time delay lines (TTDL), delay-bandwidth product (DBW), radio frequency micro-electromechanical system (RF MEMS) switch, delay variation (DV), phased arrays

## Abstract

This article presents the design method of a compact MEMS switched-line true-time delay line (TTDL) network over a wide frequency range extending from 2 to 42 GHz using TTDL units. The TTDL units, namely the cascading radio frequency micro-electromechanical system (RF MEMS) switches and GCPW, were employed in the proposed TTDL network to improve the delay-bandwidth product (DBW) while maintaining its compact size and low delay variation (DV). For comparison, a theoretical analysis of the RF MEMS switch was performed while observing the switch performance with various top electrodes. The MEMS TTDL network has a compact size of 5 mm × 5 mm, with a maximum delay of 200 ps and a minimum of 30 ps. The maximum insertion loss of 9 states is 10 dB, and the in/out return loss is better than 20 dB across 2-42 GHz. The group delay variations are within ±2.5% for all the delay states over the operating frequency range. To the best of our knowledge, the proposed TTDL network obtains the most control bits among the TTDL networks offered to date.

## 1. Introduction

In recent years, the development of true-time delay lines (TTDL) has been promoted primarily by their usefulness in replacing the traditional phase shifter in phased array systems. To improve the frequency response of the broadband phased array antenna and obtain better beam pointing characteristics, it is necessary to connect variable true-time delay lines (V-TTDL) in the antenna RF link to compensate for the effect of the antenna scanning aperture [1]. A variable true-time delay line with a fixed reference period stepping in the subarray level instead of the cell level is a standard scheme used to insert a real-time delay device in engineering [2]. At present, the main types of traditional time delay lines are SAW delay lines [3], optical fiber delay lines [4], and transmission line delay lines [5]. These devices are usually monolithic, aiming at reducing the die area of the integrated circuit, with switch powers ranging from roughly milliwatts to several watts [6].

There are many different designs for transmission-line delay lines. Some of the more prominent designs are the distributed micro-electromechanical system transmission lines (DMTLs), the reflection-type true-time delay lines, and switch-linearity true-time delay lines (TTDL). Distributed micro-electromechanical system transmission lines are well-established design methods that maintain good performance at relatively high frequencies but are generally more extensive because they contain a large number of MEMS structures. In a specific range, the higher the number of bits of the real delay Lines (TTDL), the smaller the drift of the scanning beam will be [7]. However, the published literature and the TTDL network products that have emerged generally have problems such as fewer control bits, a narrow working frequency band, and a small group delay amount. For the first time, M. Kim and J.B. Hacker of Rockwell sciences have successfully manufactured two four-bit TTD networks with a direct metal-to-metal contact MEMS switch. The TTD network was designed to produce a flat delay time over a DC-40 GHz bandwidth. However, the delay range is only 106.9~193.9 ps, with only four control bits [8]. Shiban K Koul et al. proposed a three-bit phase shifter based on a radio frequency micro-electromechanical system (RFMEMS). The phase shifter used a MEMS single-pole eight-throw (SP8T) switch. The device was fabricated on a 635 μm alumina substrate using coplanar waveguide (CPW) transmission lines. A symmetrical and compact SP8T switch is the main component of the three-bit phase shifter. The phase shifter provides an average return loss of better than 14 dB and an average insertion loss of 4.4 dB at 34.75–35.25 GHz. The mean phase error measured at 35 GHz is less than 0.98 degrees. The total area of the manufactured three-displacement phase apparatus is 5.95 mm^2^. The delay range available in the 34.75–35.25 GHz operating frequency band is approximately 80 ps to 157.8 ps [9].

This paper is organized as follows. We first recall the RF MEMS switch theory foundations while proposing a single-ended cantilever metal contact RF MEMS switch [10] model with excellent performance in DC~50 GHz. We put forward a model for a true-time delay unit in the TTDL network, which consists of a tee junction and three RF MEMS switches loading a GCPW line. Then, a TTDL network, which consists of a T-junction and three metal-to-metal contact RF MEMS switches loading a GCPW, is proposed to solve the problems mentioned above associated with existing TTDL networks. Finally, the manufacturing process used to build the switch model of the TTDL network is introduced.

## 2. Theory and Design

### 2.1. Design of the RF MEMS Switch

RF MEMS switches have superior RF performance in terms of insertion loss, isolation, power handling, and linearity over other RF switching devices [10]. To date, past advances in metal-to-metal contact MEMS switches have made the monolithic fabrication of high-quality switched-wire true-time delay (TTD) networks a relatively easy task. The structure diagram of the RF MEMS switch is shown in Figure 1.

Since a certain number of RF MEMS switches are used in the true-time delay lines, while it is very susceptible to the time delay, the switching speed must be emphatically considered. The switching time ($t_{od}$) for an over-damped system such as an RF MEMS switch *Q* < 0.5 is given [11] by
(1)$$t_{od}=\frac{2bg^{3}}{3\varepsilon_{0}AV_{S}^{2}}=\frac{9V_{P}^{2}}{8\pi fQV_{S}^{2}}$$
where b and A are the air damping constant and the top electrode area, respectively. $V_{p}$ and $V_{s}$ are the pull-in and supplied voltage. *f* and $\varepsilon_{0}$ are the mechanical resonance frequency of the switch and the permittivity of free space. The actuation voltage $V_{s}$ is selected to be slightly higher than the pull-in voltage calculated in (1), to increase the switching speed.

Four types of RF switches with different top electrodes are compared, which are a straight plate, square-etch hole, circular-etch hole, and slot-etch hole. The dimensions of the four upper electrodes are shown in Table 1. The insertion loss and voltage standing wave ratio (VSWR) are simulated using the ANSOFT HFSS software. The simulated performance is shown in Figure 2a–c.

As shown in Figure 2, the simulated insertion loss of the four top electrode switches is almost the same, but the VSWR, which significantly affects the overall matching of the true-time delay lines, has different performance. The VSWR of the switch with the straight top electrode is the worst, and the other three with etch holes are better. We can conclude that the etch hole of the top electrode can not only help to increase the switching speed but also improves the voltage standing wave ratio. As shown in Figure 2c, we use HFSS to simulate the VSWR of the full TTDL network. It can be seen that the VSWR with a circular upper electrode release hole is significantly better than that with a slot-type release hole, even though the slot provides a smaller VSWR in the case of a single switch.

### 2.2. TTDL Unit Structure and Design

A one-bit TTDL unit is shown in Figure 3a. It consists of two T-junctions, corners, two transmission lines of different electrical lengths, and three RF MEMS switches. The two T-junctions and the three RF MEMS switches connected to them can form two valid paths to select delay transmission lines with different electrical lengths, so that the signal can achieve the purpose of delaying. The “ON” state equivalent circuit diagram of a one-bit TTDL unit is shown in Figure 3b.

$Z_{0}$ is the equivalent impedance of the coplanar waveguide (CPW), $R_{c}$ is the contact resistance between the upper electrode and the CPW signal line, $C_{p}$ is the coupling capacitance at the input and output ends of the CPW signal line, $Z_{h}$ is the characteristic impedance of the upper electrode section of the RF MEMS switch, $C_{b}$ is the grounding capacitance between the air bridge membrane structure and the lower signal line, and $Z_{t}$ is the equivalent impedance of the transmission line, which contributes the group delay.

In normal conditions, the distance between the input and output of the signal line is much greater than the distance between the top electrode and the CPW signal line. Therefore, when analyzing the equivalent circuit of the RF MEMS switch, the influence of $C_{p}$ can be ignored [12]. The inductance matching section will complement the grounding capacitance $C_{p}$.

The performance of the TTDL unit is highly dependent upon the design of the T-junction, since the signal must traverse 18 switches and 18 tee junctions for the selected most significant bit (MSB) state. The T-junctions are critical components that control the flow of the RF signal, and the presence of a discontinuity in the ground of the CPW here has a significant impact on the performance of the device. Consequently, an air bridge must be erected at the corner of the T-junction to connect the ground wires on both sides of the signal line to suppress the influence caused by the asymmetry of the ground wires. However, this three-dimensional bridge membrane structure will form an additional grounding capacitance with the lower signal line, so an inductance matching section should be added [13].

The corner is a necessary part of the TTDL network to change the signal transmission direction. The CPW discontinuity also occurs at the corner because there is a significant distance difference between the electromagnetic waves propagating in the slots on both sides of the signal line, which causes odd mode transmission [14], so a transition structure is required.

By adding the inductance matching section and corner transition structure and introducing an air bridge, the insertion loss and return loss of the TTDL unit can be greatly improved, as shown in Figure 4a,b, respectively.

### 2.3. Design of TTDL Network

In engineering, group delay is always regarded as the envelope delay of a signal [15,16,17,18]. In most cases, the TTDL network needs an architecture with a specific gain and linear phase in the required frequency range to provide the best possible impedance match and delay flatness.

Equation (2) shows the relationship between the transmission line length, the equivalent dielectric constant of the substrate, and the group delay amount:(2)$$\tau_{d}=\frac{d\phi \left(\omega \right)}{d\omega }=\frac{l}{v_{p}}=\frac{l}{c}\sqrt{\frac{\varepsilon_{s}+1}{2}}$$
where $\tau_{d}$ is the group delay, l is the active length of the transmission line, $v_{p}$ is the phase velocity, $\varepsilon_{s}$ is the equivalent dielectric constant of the substrate, and c is the speed of light in a vacuum.

Based on Equation (2) and the TTDL unit pre-simulation and analysis in the previous section, we designed a 9-bit TTDL network. The MEMS switch-line TTDL network comprises nine cascaded TTDL units, using a network of GCPW lines, 25 μm wide, on a 300-μm-thick silicon dioxide substrate, and the characteristic impedance of its input/output terminals is 50 Ω. The 9-bit TTDL network was fabricated using MEMS technology and is mainly composed of a metal structure on the surface, a ground plane on the back, a through-hole between the two, and a substrate, as shown in Figure 5.

The structural parameters are shown in Table 2.

By controlling the length of the delay transmission line and optimizing the spacing between TTDL units, a TTDL network with excellent RF performance can be obtained. The structural parameters are shown in Table 1. The RF performance of the TTDL network was modeled and simulated using the ANSOFT HFSS software.

Figure 6a–d show the simulation results for the IL(S_21_), group delay, group delay error, RL(S11), and VSWR of the TTDL network, respectively. When upper switches 1~9 are closed and other switches are disconnected, it represents the start bit of the TTDL network, and the group delay is 30.01 ps. When switch 1 is off, and upper switches 2~9 and lower switches 1′~9″ are closed, it means that the first bit is enabled, the group delay is 48.04 ps, and the step is 18 ps. By analogy, from the second to ninth bits, the TTDL network has ten delay states, realizing a nine-bit controllable group delay from 30 ps to 200 ps, and the interval is 18 ps.

Table 3 shows the RF performance of the TTDL network in each state.

The simulation results show that the group delay of the most significant bit (MSB) state of the TTDL network is 193.33 ps at 42 GHz, which corresponds to a maximum insertion loss of approximately −10.07 dB. The least significant bit (LSB) state of the TTDL network is 30.01 ps at 42 GHz, which corresponds to a minimum insertion loss of approximately −1.55 dB. At 42 GHz, the difference between the insertion loss and return loss is greater than 10 dB for nine states. The group DV across the bandwidth is within ±2.5% for all the delay states. As a result, the TTDL can support wide steering angles for the phased array antenna with a broad bandwidth. This nine-bit TTDL network performs well in an adjustable TTDL network, and it can switch ten channels with fewer switches.

## 3. Comparison and Discussion

The modeled results of the MEMS TTDL network are shown in Table 4, together with other published results. DBW is defined as the product of the maximum delay and bandwidth [19]. It can be seen that this work achieves the most control bits and largest DBW.

## 4. Conclusions

A compact MEMS wideband switched-line TTDL network operating over 2-42 GHz is presented using compact TTDL units. The proposed TTDL units are composed of the RF MEMS switch and GCPW to compensate for the DV. The TTDL network provides nine-bit delay control with an MSB value of 200 ps and an LSB value of 30 ps, with an RMS delay of fewer than 5 ps, respectively. Compared with other reported TTDL networks, the proposed TTDL network has the most control bits and highest DBW to date. The results show that the proposed TTDL network is an attractive option for a wideband phased array antenna system.

## Figures and Tables

**Figure 1 micromachines-14-00246-f001:**
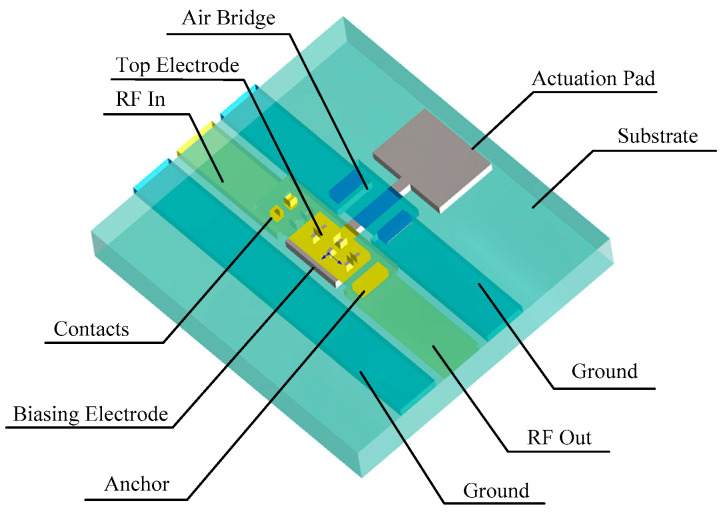
Structure of the RF MEMS switch.

**Figure 2 micromachines-14-00246-f002:**
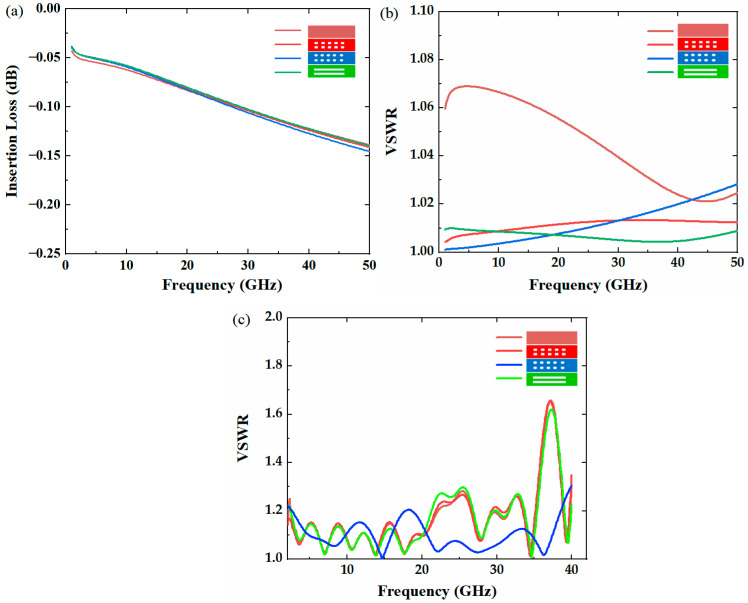
(**a**) Insertion loss (S_21_) simulation results of four different types of top electrode switches. (**b**) Voltage standing wave ratio (VSWR) simulation results of four different types of top electrode switches. (**c**) VSWR of four release holes in State_3.

**Figure 3 micromachines-14-00246-f003:**
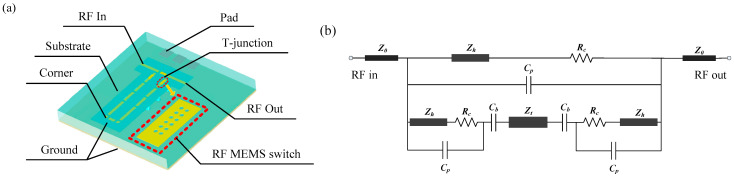
(**a**) Structure of the TTDL unit. (**b**) The “ON” state equivalent circuit diagram of the TTDL unit.

**Figure 4 micromachines-14-00246-f004:**
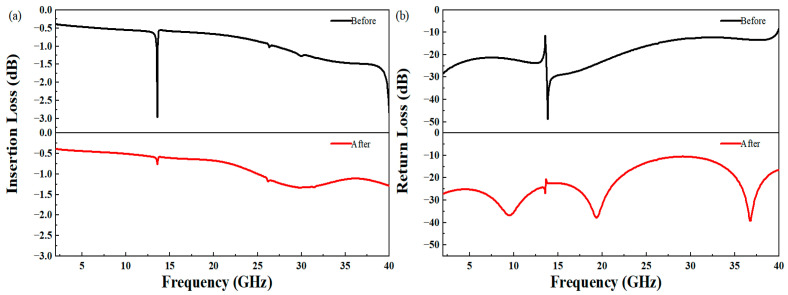
Simulation results before and after optimization for the TTDL unit: (**a**) insertion loss (S_21_), (**b**) return loss (S_11_).

**Figure 5 micromachines-14-00246-f005:**
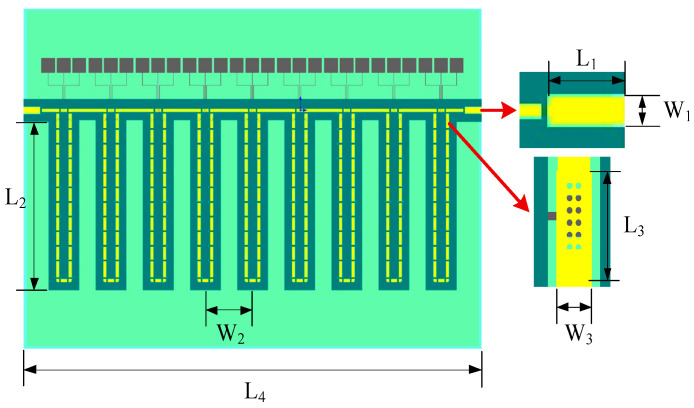
MEMS TTDL network diagram.

**Figure 6 micromachines-14-00246-f006:**
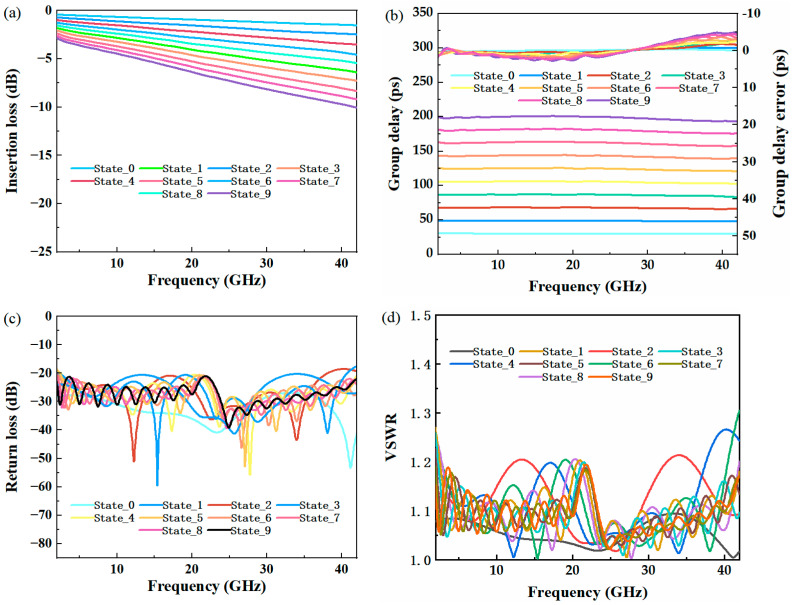
Simulated results of the TTDL network: (**a**) insertion loss, (**b**) group delay and group delay error, (**c**) return loss, and (**d**) VSWR.

**Table 1 micromachines-14-00246-t001:** Structural parameters of four top electrodes.

Dimension	Value (μm)	Comments
L	62	Length of electrode
W	25	Width of electrode
R	2	Radius of circular-etch hole
S	4	Side length of square-etch hole
S1	4	Side length of slot-etch hole
S2	41	Width of slot-etch hole

**Table 2 micromachines-14-00246-t002:** Structural parameters of the TTDL network.

Dimension	Value (μm)	Comments
L1	182	Length of port signal line
L2	3376	Length of TTDL per unit
L3	62	Length of RF switch
L4	5000	Length of substrate
W1	120	Width of port signal line
W2	515	Width of TTDL unit
W3	25	Width of RF switch
T1	300	Thickness of substrate
T2	3	Thickness of background
T3	2	Thickness of structure
D1	4	Diameter of etch hole

**Table 3 micromachines-14-00246-t003:** Corresponding TTDL network performance of each switch state (@42GHz).

State	Group Delay (ps)	Insertion Loss (I.L) (dB)	Return Loss (R.L) (dB)	VSWR
State_0	30.01	−1.55	−41.18	1.02
State_1	48.04	−2.50	−26.98	1.09
State_2	65.91	−3.54	−19.30	1.24
State_3	83.03	−4.58	−17.55	1.31
State_4	102.26	−5.47	−20.77	1.20
State_5	120.62	−6.42	−21.65	1.18
State_6	139.22	−7.30	−26.74	1.10
State_7	157.85	−8.35	−24.08	1.13
State_8	175.55	−9.21	−22.14	1.17
State_9	193.33	−10.07	−22.04	1.17

**Table 4 micromachines-14-00246-t004:** Comparison of TTDL networks.

Ref.	Frequency (GHz)	Bit (Resolution)	Maximum Delay (ps)	DBW	Technology	Size (mm^2^)
2003 [8]	0−40	3	61−86	3.44	MEMS	0.84 × 0.62
2013 [20]	15−40	3	42	1.05	CMOS	1.1 × 0.9
2014 [21]	10−50	Continuous	32.8	1.31	CMOS	0.22
2015 [22]	1−2.5	−	550	0.83	CMOS	0.07
2018 [23]	6−18	8	255	3.06	GaAs HBT	3.5 × 3.7
2020 [24]	0.1−0.5	−	2000	0.8	CMOS	0.12
2022 [25]	0−0.8	4	3800	3.04	CMOS	1.98
This work	2−42	9	200	8	MEMS	5 × 5

## Data Availability

Not applicable.

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
