# Peer review of "The Design of a Novel 2-42 GHz MEMS True-Time Delay Network for Wideband Phased Array Systems"

_micromachines, 2023, doi:10.3390/mi14020246_

Round 1

Reviewer 1 Report (Previous Reviewer 3)

The authors have addressed the concern of this reviewer. There are no additional comments, thanks.

Author Response

Dear Reviewer 1,
    We would like to thank you for your valuable comments, which are very professional and have greatly helped us to improve the manuscript.
Sincerely yours,
All authors

Reviewer 2 Report (Previous Reviewer 2)

Author Response

Dear Reviewer 2,

    We really appreciate your valuable comments, which are very professional and helpful for improving our manuscript. Following these comments, we have revised our manuscript carefully. Select 'Simple markup' in 'Revisions mode' to see all changes in Revised Manuscript. The point-by-point responses to Reviewer 1 comments are listed below. We expect that these responses provide satisfactory explanations to all concerns raised by Reviewer 2. 

    Thanks again for the reviewer’s valuable comments.

Sincerely yours,

All authors

Reviewer 3 Report (New Reviewer)

In this paper, the authors have proposed a wideband switched-line TTDL network operating over a high-frequency range [2-42] GHz. The design is using s compact TTDL units. The manuscript fits the scope of the journal, however, there few concerns that need to be addressed carefully:

A list of the proposed RF MEMS switch shown in Fig 1 is not mentioned in the paper.

Sensitivity analysis on switching time is required. 

New resonance peaks appear using TTDL for Return loss (S11). Any fundamental justifications? 

These studies are based on FEM simulations, would it be possible to cover any validations. This will strengthen the quality and show the visibility of the proposed design. 

Author Response

Dear Reviewer 3,

    We really appreciate your valuable comments, which are very professional and helpful for improving our manuscript. Following these comments, we have revised our manuscript carefully. Select 'Simple markup' in 'Revisions mode' to see all changes in Revised Manuscript. The point-by-point responses to Reviewer 2 comments are listed below. We expect that these responses provide satisfactory explanations to all concerns raised by Reviewer 3.

    Thanks again for the reviewer’s valuable comments.

Sincerely yours,

All authors

Round 2

Reviewer 3 Report (New Reviewer)

Thanks for the updated version.

This manuscript is a resubmission of an earlier submission. The following is a list of the peer review reports and author responses from that submission.

Round 1

Reviewer 1 Report

This work describes a simulated design for a true-time delay line (TTDL) that achieves a high maximum delay over a wide bandwidth.  It also includes fabrication procedures and measurements of a switch that could be used to implement the simulated TTDL.  I raise two technical issues and offer and a few style suggestions below.

The first technical issue is about a claim (on line 96) that circular etch holes are chosen because they maximize linearity of VSWR.  Why is it desirable for VSWR to be linear with frequency?  Please explain further.  I would expect that the lower VSWR offered by square holes or slots would be more desirable.

The second technical issue is about the results reported in section 4.2, which have a few issues:

·       First, I think these results are intended as a proof-of-concept to show that the simulated design from Figure 5 could be built, but the insertion loss of this switch is nearly 3dB, much more than earlier simulations.  For instance, figure 2 shows only 0.15dB insertion loss in the simulated switch, and Figure 4 shows 1.5dB of loss for the whole unit delay line structure. 

·       Second, the switching action described in lines 241-244 cannot be observed in Figure 11.  Lines 241-244 say that the switch achieves stable contact after multiple bounces, but Figure 11 just shows a capacitive charging behavior, which raises a variety of questions:

o   Is channel 1 connected to the switch gate and channel 2 connected to RF Out?

o   What is RF Out loaded by in this experiment?

o   Is the figure just showing the switch charging the capacitance of the probe?

o   Where can I infer the switching delay from the figure?

o   Where can I observe bouncing in the figure?

·       Third, though reliability is mentioned in the text, there are no reported measures of reliability, only text assurances that “the repeatability is good”.  Figures showing waveforms after 10^6 cycles would be more convincing, as would some measured cycles-to-failure numbers. 

·       Finally, this measured data is something of a bait-and-switch because the preceding paper was about the full TTDL.  The text needs to be diligent about explaining why a single switch measurement fits in a paper about a simulated TTDL.  Obviously, a full experimental demonstration of the TTDL would be more convincing than demonstrating a single switch, but such a measurement is probably outside the scope of this paper. 

A substantial rewrite of the experimental results, expanding the experimental details, clarifying their relation to the simulations and improving the measured traces in Figure 11 would improve the paper.

In addition to these technical issues, I have a number of style notes.  (Though it is worth noting that I enjoyed reading this paper: the text is thoughtfully composed and clearly written.)

·       Lines 106-107 – The caption for figure 3 is separated from the figure by a page break.  The caption must be on the same page as the figure.

·       Line 108 – This sentence describes variables in Figure 3b without ever referring to figure 3b.  That connection needs to be made clear in the text.

·       Lines 155-157 – describe a technology stackup and say that stackup is pictured in Figure 5.  No such stackup appears in figure 5, which is just an isometric rendering of the whole TTDL structure.  Moving the technology stackup from Figure 7 to here would make more sense with the text. 

·       Line 165 – Use “delay line” or “delay line unit” instead of the ambiguous “delayer”.

·       Line 165 – This is the first mention of HFSS even though we’ve been sharing simulated results since figure 2.  The methodology should be introduced where it is first used.

·       Table 1 – These dimensions don’t appear on any figure.  A layout figure with these dimensions annotated should be included.

·       Figure 6 (right) – the left axis of this figure is mislabeled.  It says “Return Loss” and I think it should say “Group delay”.  In general, I think the left and right axes in figure 6 are doing more harm than good, and I think four charts would be easier to read than the current version of the plot.

·       Figures 8 and 9 – should be centered.  Could possibly be merged into one figure.

·       Throughout – figure/table to text spacing is inconsistent.  There should be some space.

·       Line 245 – typo: reference [255] should probably be reference [25]

·       Line 247 – possible typo: I think “10^6@” should be “10^6”, the @ sign should be deleted.

Author Response

Dear Reviewer 1,

    We really appreciate your valuable comments, which are very professional and helpful for improving our manuscript. Following these comments, we have revised our manuscript carefully. Select 'Simple markup' in 'Revisions mode' to see all changes in Revised Manuscript. The point-by-point responses to your comments are listed below. We expect that these responses provide satisfactory explanations to all concerns raised by you. 

    Thanks again for your valuable comments.

Sincerely yours,

All authors

Reviewer 2 Report

The paper is well organized and presents a RF MEMS-based true-time delay network. The proposed circuit is quite large and many issues remain.

-      - Authors need to improve the introduction section through adding a review study of recently reported MEMS-based true time delay circuits in more detail. Also, explain the problems associated with the prior art and the novelty of your research work.

-          - The sentence in line 81 is not clear!!!

-          - It is not clear where do the values in Fig. 2 come from. Authors could add a table showing the insertion loss and VSWR equations.

-          - The effects of non-ideality and mismatch of the TTDL unit should be discussed.

-          - “A one-bit TTDL unit…” has been repeated in line 103 (refer to line 98).

-         - In Fig. 3(b) there are contact resistances Rc, but you mentioned Rs in line 108.

-          - Axis label in Fig. 6(b) should be Group Delay (ps).

-         - In Table 3, authors need to compare more recently reported circuits in the same technology. In phased array systems, delay resolution is more important than bit resolution. Provide in Table 3: delay resolution, delay range, and gain (average gain). Also, mention the references are based on simulation or measurement. What is the noise figure of the TTDL?

-          - (a) and (b) are not inserted in Figs 8 and 9.

-          - “Figs 11(a) and (b) show…” in line 241.

-          - Correct the reference provided in line 245.

Author Response

Dear Reviewer 2,

We really appreciate your valuable comments, which are very professional and helpful for improving our manuscript. Following these comments, we have revised our manuscript carefully. Select 'Simple markup' in 'Revisions mode' to see all changes in Revised Manuscript. The point-by-point responses to your comments are listed below. We expect that these responses provide satisfactory explanations to all concerns raised by you.

Reviewer 3 Report

This manuscript presents a wide-band 2-40 GHz delay line with MEMS technology with a simulated maximum delay of 200 ps. The performance of the results is quite impressive. There are a few comments to the Author: 1.      Table 2 is a little bit confusing. It shows bit 0 to bit 9 states, which implies that Bit 0+ Bit 1+…+Bit 9 can be added to get more delay. It is suggested to put State0, State9, etc, and make it more transparent. Besides, the comparison to the conventional switched line bit distribution should have been included. 2.      Table 2 is NOT complete, VSWR should have been included. The comparison table is NOT complete, Insertion loss can be included. 3.      Power handling capability can be added. The actuation voltage should also be listed as a critical factor of the RF MEMS switch.

Author Response

Dear Reviewer 3,

We really appreciate your valuable comments, which are very professional and helpful for improving our manuscript. Following these comments, we have revised our manuscript carefully. Select 'Simple markup' in 'Revisions mode' to see all changes in Revised Manuscript. The point-by-point responses to your comments are listed below. We expect that these responses provide satisfactory explanations to all concerns raised by you.

    Thanks again for the reviewer’s valuable comments.

Sincerely yours,

All authors

Round 2

Reviewer 1 Report

This work describes a simulated design for a true-time delay line that achieves a high maximum delay over a wide bandwidth.  It also presents fabrications procedures and measurements of a RF switch that could be used in the design.  In my last review report I requested the following changes:

1) An explanation of the choice to use to use circular release holes since their VSWR seemed higher than other release hole options.  This explanation was to replace the claim that the hole shape was picked because it improved the “voltage standing wave ratio linearity”.

2) Improved explanations of the switching data in Figure 11, accompanied at the authors’ discretion by an improved figure, that clearly explained the switch’s electrical operation, illustrated bouncing and highlighted the extraction of mechanical delay.

3) Experimental evidence of the reliability performance of the switch.

4) Improvement of a number of style and organization issues, notably including revisions to figure 5 and table 1, and improvements in the explanation linking the experimental data to the earlier simulations.

The authors responded to these requests by providing a letter answering my questions and by modifying the manuscript in the second revision.  However, though these requests have been addressed in the response letter, only request 3 is addressed in the revised manuscript.   I discuss the status of each request below and clarify the changes that are required to satisfy them.

During review I also noticed that several pieces of device information that are needed to assess the switch behavior are missing.  This leads to a new request 5 – what is the pull-in voltage of the fabricated device, and what is the actuation gap?

Request 1 – VSWR discussion

This request is fully addressed in the response letter by including a VSWR simulation for a full TTDL structure that indicates why circular release holes were used.  However, neither this figure nor the accompanying explanation of its results appears in the revised manuscript.  The following changes to the manuscript would satisfy this request:

1a) The simulation of the full TTDL should be included in the manuscript and used to justify the choice of circular release holes for lower VSWR. 

1b) The simulation should be accompanied by explanatory text similar to the text in the response letter. 

1c) The clause on lines 96-97 that reads, “the voltage standing wave ratio linearity with the circular etching hole on the upper electrode is the best”, should be removed and replaced with an explanation that the VSWR of a full TTDL is lowest with circular etch holes, even though other etch hole shapes have lower VSWR for a single switch.

Request 2 – Measured Insertion Loss and Switching Data in Figure 11

Though the authors speak to this request in the review letter, I am left with further questions about the data.  The discussions in the review letter are also not reflected in the manuscript, where the text describing Figure 11 is largely unchanged.

First, the response letter discusses differences in insertion loss between different simulated and measured figures, attributing them to different lengths of waveguides attached to switches.  If that is the case, provide the (ideally measured) loss per meter of a waveguide across your frequency range, and annotate each insertion loss measurement with the length of attached waveguide.  This is important information for the readers to assess the differences between figures 2, 4, 6, 10 and the text in section 4.2 for themselves.

Next, the response letter discusses my concerns about figure 11, but I find the response unsatisfactory.  It is unsatisfactory for two reasons: figure 11 still does not sufficiently describe the electrical test being performed, and I still don’t see evidence of bouncing or mechanical delay in figure 11.

Regarding the description of the electrical test: there are several figure-issues, which are listed below.

2a) Neither the response letter nor the manuscript provides any information about the source impedance driving this switch or the load impedance.  Because the data is an oscilloscope trace, I assume the load is an oscilloscope probe impedance (usually dominated by a ~10pF capacitor).  The capacitance of the cabling may also play into the measured response.  The source impedance is probably 50 ohms, but it’s hard to tell.  Notably, the oscilloscope used in this measurement is not listed in the equipment described in lines 231-232.  This work must provide information about the loading condition of the switch under test.

2b) the response letter says that CH1 of the oscilloscope is the drive signal, which I presume is the gate voltage and not RF IN, and that CH2 is the response signal, which I presume is RF OUT.  This description is not included in the manuscript, and figure 11a does not have a measured CH1 signal, but instead shows an R1 signal.  Exporting the data and plotting it with a proper legend, then describing the figure in the text would fix this figure-issue.  Be sure to disambiguate between the gate drive and RF IN in the description.

2c) The response letter claims that because the CH2 signal is high when the CH1 signal is high, and because the CH2 signal is low when the CH1 signal is low, the illustrated behavior must be a switch.  That explanation is insufficient because the same behavior would also occur in a charging and discharging RC network. Figure 11 shows a curve that obeys either first order or overdamped second order dynammics.  If this curve represents the position of the device, then the author’s claims about the switching behavior are accurate, but there is no indication that vibrometry is being used to measure the position of the switch.  If this curve represents the RF OUT voltage, which matches the description in the response letter, then the first order part of the curve can’t describe switching behavior, and instead must represent charging a load capacitance.  An RF OUT measurement can be used to indicate switching behavior, but not at the level of zoom used in the figure.  Switching behavior would look like a delay between the rising edge of the gate signal and the start of a RC response in the RF OUT signal. The mechanical switch would be in transit during that time, which would prevent it from charging or discharging the load.  This figure must indicate the mechanical switching delay, probably by zooming in on the transition of the gate.

2d) My concerns that the measured curves provide no evidence of bouncing are not addressed in either the response letter or the manuscript.  The simplest way to address these concerns is to remove the claim that there are multiple bounces from line 241.

Finally, the response letter clarifies that the switch demonstration is intended to show that the most difficult step in TTDL fabrication can be overcome.  That claim needs to be carefully couched and expanded on as discussed further under the “Request 4” section below.

Request 3 – Reliability Data

Data was provided in the response letter and incorporated into the manuscript.  I commend the authors for gathering and adding this data.

However, the text explaining these new reliability figures is cursory, and the figures themselves violate some crucial style principles, and there is a faulty claim about input and output duty cycles that is made in the response letter. This request will be satisfied if the authors make the following changes to improve the explanation of these plots:

3a) Include time and voltage axes on all plots.  All plots must have labeled x and y axes.

3b) Include the gate voltage and RF IN signals, and include a legend indicating which signal is which.  I presume that we’re looking at RF OUT signals in this plot, but these waveforms could be created by a switching gate voltage and a DC RFIN, or they could be created by a switching RF IN and a DC gate voltage.  Including and labeling all of the signals on each scope trace would clarify what is happening.  It is also acceptable to include only the varying signal, and to describe the DC signal in text and caption.

3c)   Label each plot with the number of cycles that have elapsed, the power applied during each cycle, and a device ID.  It’s not clear to me if figures 12c and 12d are further deteriorations of the behavior observed in 12b as more cycles are applied, or if these are failure modes at the same number of cycles (and applied power levels) on different switches.

3d) Include further details in the text and captions: where these devices hot or cold switched? Was the power DC or AC?  (I presume AC because it is specified in dBm, but it’s hard to be sure.)  What was the load capacitance?  All of these factors affect cycle life of switches.

3e) Either remove the claim on line 248 that “the insertion loss changed little” or back the claim up with a measured insertion loss. Please also clarify which experiment’s insertion loss you are using to make this claim.  This clarification is necessary because insertion loss looks bad in some of your results. For example, it looks like the insertion loss in figure 3c is very high because of incomplete conduction.

3f) Amend the text so it accurately describes the figures after the above changes have been made.

3g) Don’t include the claim that a 50% gate drive signal will necessarily result in a 50% RF OUT signal in the final manuscript.  The pull-in delay and release delay of the switch are almost certainly different even without accounting for changes in surface adhesion induced by wear.  If the period of the switching signal is much longer than the pull-in or pull-out delay, then it is true that the RF OUT signal will be close to 50%, but that test condition needs to be stated in the text.

Request 4 – Style Changes

My initial review report requested many changes, and many of these changes were discussed in the response letter and incorporated into the revised manuscript.  Thanks to the authors for attending to those changes.

I’d like to focus this section on two important stylistic improvements that I believe must be implemented to the manuscript in this section: improving figure 5 and table 1, and fully explaining how the experimental results relate to the earlier simulations.  Other lesser issues also linger in the manuscript, and I include a full accounting of outstanding style issues below in the “Oustanding Minor Style Issues” section below. 

Regarding figure 5 and table 1: by the author’s own admission in their response letter, it is difficult to label the dimensions from table 1 on the current version of figure 5.  No amount of improvement in table 1 will unambiguously show how those dimensions translate to figure 5.  Replace figure 5 with a top-down view and annotate it with the dimensions from table 1.  If the top-down view is too zoomed out to effectively annotate a dimension, include a call-out or inset at the appropriate level of zoom.

Regarding the explanation of how the experimental results relate to the TTDL simulations: The first half of this paper is (mostly) about a full TTDL system, but the experimental demonstration in the paper is about a single switch.  Demonstrating a single RF switch is not a new contribution on its own, so the justification for publishing this paper rests on explaining why the switch is an important building block for the TTDL.  Such an explanation could fit naturally between lines 193 and 194.  The explanation given in the response letter – “If the performance of the RF MEMS switch meets the requirements, we think that the manufacturing of the TTD network is no longer difficult.” – is not a sufficient reason to exclude this explanation; I’ll believe the TTDL results are easily achieved when I see them measured.

When explaining the connection between the switch and the TTDL, it is important to justify the specifications you’ve selected, which you refer to for the first time on line 234.  Why is 2.7dB of insertion loss in the measured switch acceptable when the simulated loss of a whole TTDL unit is <1.5dB?  Where does this specification come from?  Naively, your measured results suggest that this switch would result in a TTDL with >27dB of loss, which seems unacceptably high.

Outstanding Minor Style issues

Lines 78-80 – the description of equation 1 does not describe what all the variables stand for.  What’s f? What’s e0?  I can guess, but the paper should say.

Lines 88-89 – the figure 2 caption should specify that these results are for a single switch, and not a full TTDL.

Line 175 – the caption of figure 6 refers to a TTD network, but the text at line 186 refers to a TTDL network.  Use a standard acronym throughout.

Line 238 – the figure 10 caption should specify that these results are for a single switch, and not a full TTDL.

Line 241 – “the switch drops from the pull-down voltage to zero” implies the existence of a pull-down resistor.  I think the paragraph should be rephrased starting at the end of line 239 “As the actuation signal rises from zero to the pull-in voltage, the switch is actuated to the contacting position and achieves a stable contact.  When the actuation signal drops from the pull-in voltage to zero, the switch is released from the contacting position and returns to its resting position.”  Per comments elsewhere in this review, I also think this paragraph would further benefit from describing how the position of the relay is reflected in figure 11.

Line 247 – instead of “10^6@10dBm times” say “10^6 cycles while carrying 10dBm of power”

Author Response

Dear Reviewer 1,

    We really appreciate your valuable comments, which are very professional and helpful for improving our manuscript. Following these comments, we have revised our manuscript carefully. Select 'Simple markup' in 'Revisions mode' to see all changes in Revised Manuscript. The point-by-point responses to Reviewer 1 comments are listed below. We expect that these responses provide satisfactory explanations to all concerns raised by Reviewer 1. 

    Thanks again for your valuable comments.

Sincerely yours,

All authors

Round 3

Reviewer 1 Report

The author’s response to the latest draft was not attached.  However, the authors have made substantial headway in addressing the issues that I have raised in the manuscript.  I discussed five requests for changes in my last review report:

1) An explanation of the choice to use to use circular release holes since their VSWR seemed higher than other release hole options.  This explanation was to replace the claim that the hole shape was picked because it improved the “voltage standing wave ratio linearity”.

2) Improved explanations of the switching data in Figure 11, accompanied at the authors’ discretion by an improved figure, that clearly explained the switch’s electrical operation, illustrated bouncing and highlighted the extraction of mechanical delay.

3) Experimental evidence of the reliability performance of the switch.

4) Improvement of a number of style and organization issues, notably including revisions to figure 5 and table 1, and improvements in the explanation linking the experimental data to the earlier simulations.

5) Inclusion of pull-in voltage and gap separation of the switching device.

The authors have fully addressed issues 1 and 4 in this latest revision of the manuscript.  They have partially addressed issue 3 as well by including a time scale on figure 12 and amending the text, but questions remain about the relationship between subfigures 12b, 12c and 12d, whether the device is hot switched with a DC analog signal (the signal comes from a signal generator and is described as 10dBm, but RF IN appears constant in Figure 12 if the actuation signal matches the description in the text) and the time constant and resistance of the load. 

Requests 2 and 5 remain unaddressed (though the unsupported claims about bouncing have been removed), and the failure to address request 2 is a critical issue with the presentation of data in this paper.  I don’t see evidence of mechanical switching in Figure 11, and I think the authors need to provide that evidence to match their claims about the operation of their switch.  The present form of figure 11 just shows a capacitive charging behavior, which is a property of the switch load instead of the switch mechanical behavior.  As a result, I think that the author’s claimed on/off switching time is a measure of the charging or discharging of a load network, not the mechanical behavior of a switch.

(It is worth nothing that figure 12 does show behavior consistent with wear on a mechanical switch, but the time scale in that figure is also too long to observe switching dynamics, and the figure is further limited because it does not show an input signal for comparison.)

I refer the authors to my second review report for details of how to address issues 2, 3 and 5, since my requests remain essentially unchanged.  As a minimal summary, the authors must address issue 2 by providing evidence of mechanical switching, which will require showing a much shorter time scale on figure 11 around the rising edge of the actuation signal.  Request 3 can be addressed by describing whether the RF IN signal is AC or DC, whether the switches are hot switched, the resistance and capacitance of the envelope detector load, and by explaining whether figures 12b, 12c and 12d come from the same device or from different devices.  Request 5 can be addressed by including the thickness of the sacrificial polyimide described on lines 228 and 233 and the switch pull-in voltage.

Minor style issues:

Line 261 – The configuration of the device under test, the test equipment, and the envelope detector would be best indicated by a schematic. 

Line 267 – “exceeds the limit after 106” should be “exceeds the limit after 10^6 cycles”